# Changes of Air Pollution between Countries Because of Lockdowns to Face COVID-19 Pandemic

**Aytac Perihan Akan** [1] and **Mario Coccia** [2,*]

1 Department of Environmental Engineering, Hacettepe University, 06800 Ankara, Turkey
2 Department of Social Sciences and Humanities, CNR—National Research Council of Italy, 10135 Torino, Italy
* Correspondence: mario.coccia@cnr.it

**Abstract:** The goal of this study is to analyze how levels of air pollution changed between countries with their restriction policy of lockdown to cope with the COVID-19 pandemic. The study design compares average changes of $CO$, $NO_2$, $SO_2$, $O_3$, $PM_{2.5}$ and $PM_{10}$ concentrations based on measurements at ground level in January, February, and March for the years 2019, 2020, 2021, and 2022 (during the COVID-19 pandemic crisis) to average values of a 2015–2018 baseline period (ex-ante COVID-19 pandemic) between 300 cities in 19 countries of five geoeconomic regions. Results reveal that the maximum reduction in air pollutant concentrations is given by: $CO$ ($-4367.5\%$) in France, $NO_2$ ($-150.5\%$) in China and Australia, $SO_2$ ($-154.1\%$) in Israel, $O_3$ ($-94.1\%$) in China, $PM_{2.5}$ ($-41.4\%$) in Germany, and $PM_{10}$ ($-157.4\%$) in Turkey. Findings show that the effects of restriction policies for COVID-19 pandemic on air quality vary significantly between countries, depending on the different geographical, economic, industrial and social characteristics of the countries. These results clarify the critical relationship between control measures for pandemic crises and levels of air pollution in countries that can support best practices of environmental policy for pathways of sustainable development.

**Keywords:** COVID-19 pandemic; restriction policies; lockdown; air pollution; atmospheric pollution; air pollutants; air quality monitoring; air quality; environmental policy; climate change





## 1. Introduction

The COVID-19 pandemic crisis started in 2019 with the novel viral agent SARS-CoV-2, generating many socioeconomic and health issues worldwide and affecting societal behavior, institutions, and also the environment [1–3]. The intensity of COVID-19 waves, driven by different variants (e.g., Delta, Omicron, etc.), has generated adverse impacts on the health of people in terms of deaths and infected people both in developed and developing countries [4–9]. The transmission forces of COVID-19 are driven by various factors, such as the high density of cities, high level of air pollution, intensive commercial trade, atmospheric stability with low wind speed, etc. [10–17]. In the presence of the COVID-19 pandemic crisis, governments implemented non-pharmaceutical measures of control, such as contact tracing systems, partial or full lockdown measures on a regional or national scale, etc., especially during the peaks of the COVID-19 outbreaks [15,18–23]. In the first wave of the COVID-19 pandemic, the SARS-CoV-2 virus was spreading rapidly between crowded human groups, and lockdown restrictions were the best option to cease or decelerate the transmission dynamics of the novel virus from human to human [19]. Restriction policies based on full lockdown and quarantine have generated a negative impact on countries' economies, but they have also reduced levels of air pollution nationally and regionally [19,24–28]. After China, the first countries to apply a partial lockdown were Italy and Iran because the number of confirmed cases and deaths of COVID-19 increased rapidly [15,19]. In the year 2020, partial or full lockdowns were carried out in many nations worldwide [19,29].

Manifold studies show the effects of lockdown measures on levels of air pollutants and air quality in countries. Li et al. [30] determined the effects of the quarantine on air quality in the Yangtze River Delta region of Eastern China. Results suggest remarkable declines in the concentrations of air pollutants from industry and traffic emissions because of the diminishing of socioeconomic activities. The reduction in pollutant concentrations in 2020, which contributed considerably to the improvement of the region's air quality, was found for fine particulate matter ($PM_{2.5}$), nitrogen dioxide ($NO_2$), and sulfur dioxide ($SO_2$), when data were compared to the year of 2019 [30]

Sannino et al. [31] analyze the impact of the quarantine measures on the air pollution in the city of Naples, Italy, from 13 March to 4 May 2020, analyzing gaseous pollutants (benzene, $C_6H_6$), carbon monoxide (CO), $NO_2$ and $SO_2$ and particulate matter ($PM_{10}$, $PM_{2.5}$, $PM_1$) at ground level. Results showed that $NO_2$ was reduced by 49–62% in urban and green suburban areas, although CO and $SO_2$ indicated a higher decrease in urban or industrial regions of the city (50–58% and 70%, respectively). PM concentrations also showed a reduction ranging between 29 and 49% [31]. Another study by Benchrif et al. [32] investigated the way that levels of air pollutants ($PM_{2.5}$ and tropospheric $NO_2$) changed before, during, and after a lockdown in 21 cities in different countries. Results exhibited a drop in $NO_2$ concentrations ranging from 3 to 58% during the lockdown period, except for three cities (Abidjan, Conakry, and Chengdu), which observed an increase in the $NO_2$ levels at a rate of 1, 3 and 10%, respectively. Instead, $PM_{2.5}$ levels exhibited an increase after the lockdown period.

Bray et al. [33] analyzed the differences in air pollutants (CO, $NO_2$, $SO_2$, $O_3$, $PM_{2.5}$, and $PM_{10}$) by comparing the quarantine period of March and April 2020 to the same months in 2015 and 2019. Globally, $NO_2$ levels decreased by around 9.19% and 9.57% in March and April 2020. Regionally, most monitoring sites in Europe, the USA, China, and India displayed reductions in the concentration of air pollutants containing CO, $NO_2$, $SO_2$, $PM_{2.5}$, and $PM_{10}$ during the quarantine period, whereas only $O_3$ concentrations increased during the same period. In Valencia, Spain, Donzelli et al. [34] assessed the effects of restrictions to cope with COVID-19 on air quality and pollutant emissions, including nitrogen oxides ($NO_x$), nitric oxide (NO), $NO_2$, $PM_{10}$, $PM_{2.5}$, and $O_3$, comparing the same periods in 2020 and 2019. The highest reduction of the $PM_{10}$ and $PM_{2.5}$ levels for the València Centre, València Avd Francia, and València Pista de Silla was given at 58–42%, 56–53%, and 60–41%, respectively. Similarly, a remarkable decline in NO levels was also recorded in all air monitoring stations. In particular, $NO_x$, $NO_2$, and NO concentrations decreased in the range of 37.4–65.5%, 35.7–67.7%, and 35.3–63.5%, respectively, in 2020. Finally, $O_3$ levels were reduced during the lockdown period. Filonchyk et al. [35] analyzed the variations of air quality parameters ($PM_{2.5}$, $PM_{10}$, $NO_2$, and $SO_2$) in five large cities of Poland, comparing data of the lockdown period in 2020 to data from 2018 and 2019. They showed decreasing levels in aerosol concentrations in April and May 2020 of approximately $-23\%$ and $-18\%$ compared to the 2018–2019 period. For $PM_{2.5}$ and $PM_{10}$, the reductions were from $-11.1\%$ to $-26.4\%$ and $-8.6\%$ to $-33.9\%$ in April 2020 and from $-8.7$ to $-21.1\%$ and $-8.5\%$ to $-31.5\%$ in May 2020, as compared to the same months in 2019.

Lonati and Riva [36] examined the variations in the concentration of gaseous air pollutants containing $NO_2$, benzene, and ammonia-based during lockdown restrictions applied in the Po Valley of Northern Italy, comparing data of the COVID-19 pandemic crisis to the previous 6 years, on monthly, daily and hourly bases. Results showed improvements in air quality during the spring of 2020 because of the reductions in nitrogen oxides and benzene emissions, associated with $-50\%$ of road traffic in urban areas. Clemente et al. [37] analyzed the effects of COVID-19 measures of restriction on concentrations of $PM_1$, $PM_{10}$ and their chemical components in some urban regions of the Western Mediterranean, by comparing data during the lockdown of 2020 with the previous five years. Results revealed that the average reduction in $NO_x$ and volatile organic compounds was higher than 50%, while ground-level ozone concentrations did not display considerable differences during the study period. Moreover, a 35% drop in $PM_1$ and $PM_{10}$ levels was recorded when

Saharan dust events were excepted from the period under study. Hence, traffic limitations associated with lockdown measures contributed to significant declines in elemental carbon and heavy metal concentrations resulting from road dust. In addition, nitrate showed the largest reductions for the decline in regional emissions of $NO_x$ regarding secondary inorganic aerosols.

Cucciniello et al. [38] investigated the impact of lockdown measures in Avellino, Italy on air pollution by analyzing the concentrations of CO, $O_3$, $PM_{2.5}$, $PM_{10}$, $C_6H_6$, and $NO_2$ during the period January–December 2020. They showed significant reductions in CO, $C_6H_6$, and $NO_2$ pollutants during the examined period, in particular March 2020. In a study implemented in Kabul, Afghanistan, Himat [39] analyzed the city's air quality by examining air quality parameters containing $PM_{10}$, $PM_{2.5}$, CO, $SO_2$ $NO_2$, and $O_3$ for the pre-and post-COVID-19 period. Emission data in different regions of Kabul between 2020 and 2018 showed that emissions exceeded the standard values of 150 and 75 $\mu g/m^3$ for 24 h, especially for $PM_{2.5}$ and $PM_{10}$, ten times. The same situation has been observed in $SO_2$ concentrations; the increase is due to the high utilization of stone slag in different regions of Kabul, especially in high-rise buildings and bathrooms. Moreover, air pollution decreased in Kabul for the period of February-April 2020, with control measures for COVID-19, whereas the average concentration of $PM_{2.5}$ emissions increased in May 2020, when lockdown restrictions were suspended.

Other studies focus on changes in the air pollution level associated with the restriction policy of lockdown to face COVID-19 in different countries, such as Kutralam-Muniasamy et al. [40] for Mexico City, Mor et al. [28] for Chandigarh, located in the Indo-Gangetic plain of India, Chowdhuri et al. [41] for Kolkata Metropolitan Area, India, Das et al. [42] for Mumbai, India, and Pal et al. [43] for the Indian cities of Delhi, Mumbai, Kolkata, and Chennai. Sathe et al. [44] also analyzed some Indian cities (e.g., Mumbai, Bengaluru, and Kolkata), Wetchayont et al. [45] analyzed Bangkok Metropolitan Thailand, Jakob et al. [46] looked at Jakarta, Indonesia, Upadhyay et al. [47] investigated regions in South Asia, Gao et al. [48] investigated Wuhan, China, Celik and Gul [49] Istanbul, Turkey, Anil and Alagha [24] and Alharbi et al. [50] the Kingdom of Saudi Arabia, Sbai et al. [51] the city of Lyon and the center of the Auvergne-Rhône-Alpes region, France, Jeong et al. [52] Toronto, Canada, Gorrochategui et al. [53] Barcelona metropolitan area and other parts of Catalonia, and Skirienė et al. [3] examined the United Kingdom, Spain, France, and Sweden, as well as the Northern Italy region, etc.

Table 1 systematizes studies that analyze the effects of the control measures to cope with the spread of the SARS-CoV-2 virus per main parameter of air pollution in different periods across countries.

In general, these studies show significant reductions in the concentrations of primary air pollutants (CO, $NO_2$, $SO_2$, $PM_{2.5}$, and $PM_{10}$), one of the secondary pollutants ($O_3$) and also the concentration of $CO_2$ [54–56].

Most of the studies investigated the effects of the lockdown to face the COVID-19 pandemic on air pollution until the year 2021.

The principal goal of the paper here is to expand results of these studies in order to clarify the relationship between the containment policy of lockdown and levels of air pollution using new data up to August 2022 between different countries worldwide. In particular, this study here analyzes how levels of air pollutants, given by CO, $NO_2$, $SO_2$, $O_3$, $PM_{2.5}$, and $PM_{10}$, change with restriction policy (lockdown) across nineteen countries of five geoeconomic regions from 2015 to 2022.

**Table 1.** Studies concerning the effects of lockdown measures to cope with the COVID-19 pandemic on levels of different air pollutants.

| Parameter | Region, Country | Variation in Air Pollutants |
| --- | --- | --- |
| CO | Bogota, Colombia | 23–34% reduction [57] |
| | Quito, Ecuador | 48.75% reduction [58] |
| | Beijing-Tianjin-Hebei, China | 20.40% reduction [59] |
| | Dhaka, Bangladesh | 8.8% reduction [60] |
| | Delhi, India | 30.35% reduction [61] |
| | Nagpur, India | 63% reduction [62] |
| | USA and China | 19.28–25.53% reduction in USA and China [63] |
| | Santiago, Chile | 13% reduction [64] |
| $NO_2$ | Bogota, Colombia | 13–22% reduction [57] |
| | Delhi and Mumbai, India | 60–78% reduction for Delhi and Mumbai [65] |
| | Barcelona, Spain | 66% reduction [66] |
| | Quito, Ecuador | 63.98% reduction [58] |
| | China | $19.1 \pm 9.4\%$ reduction [67] |
| | Vietnam | 24–32% reduction [68] |
| | Nagpur, India | 69.2% reduction [69] |
| | Makkah, Saudi Arabia | 58.66% reduction [70] |
| | Beijing-Tianjin-Hebei, China | 37.80% reduction [59] |
| | Dhaka, Bangladesh | 20.4% reduction [60] |
| | Bangladesh | 40% reduction [25] |
| | United Kingdom | 38.3% reduction [71] |
| | Leeds, Sheffield, and Manchester, England | 37.13–55.54% reduction [27] |
| | Ankleshwar, Vapi and Gujarat, India | 80.18% reduction [72] |
| | Delhi, India | 52.68% reduction [61] |
| | Bengaluru (Bangalore), India | 87% reduction [62] |
| | USA and China | 36.7–38.98% reduction in USA and China [63] |
| | Rio de Janeiro and São Paulo, Brazil | 10–40% reduction [73] |
| $SO_2$ | Bogota, Colombia | 11–20% reduction [57] |
| | Delhi and Mumbai, India | 19–39% reduction for Delhi and Mumbai [65] |
| | Quito, Ecuador | 45.76% reduction [58] |
| | Nagpur, India | 64.3% reduction [69] |
| | Dhaka, Bangladesh | 17.5% reduction [60] |
| | Bangladesh | 43% reduction [25] |
| | USA and China | 3.81% increase in the USA–18.36% reduction in China [63] |
| $O_3$ | Bogota, Colombia | 31.3–14.1% increase [57] |
| | Barcelona, Spain | 27% increase [66] |
| | Quito, Ecuador | 26.54% increase [58] |
| | Makkah, Saudi Arabia | 68.67% increase [70] |
| | Morocco | 22–28% increase [74] |
| | Dhaka, Bangladesh | 9.7% reduction [60] |
| | Bangladesh | 7% increase [25] |
| | United Kingdom | 7.6% increase [71] |
| | Santiago, Chile | 63% increase [64] |
| $PM_{2.5}$ | Bogota, Colombia | 7–15% reduction [57] |
| | Quito, Ecuador | 42.17% reduction [57] |
| | Jiangsu, China | 18% reduction from pre-COVID; 2% decrease post-COVID [75] |
| | Valencia, Spain | 3.1% increase [76] |
| | Toronto, Canada | 4% reduction [52] |
| | Beijing-Tianjin-Hebei, China | 21.50% reduction [59] |
| | Leeds, Sheffield, and Manchester, England | 29.93–40.26% reduction [27] |
| | New Delhi, Chennai, Kolkata, Mumbai, and Hyderabad, India | 62% reduction, followed by Mumbai (49%), Chennai (34%), and New Delhi 26% [77] |
| | Ahmedabad, India | 68% reduction [62] |
| | Santiago, Chile | 11% reduction [64] |

**Table 1.** *Cont.*

| Parameter | Region, Country | Variation in Air Pollutants |
|---|---|---|
| $PM_{10}$ | Bogota, Colombia | 25–16% reduction [57] |
| | Delhi and Mumbai, India | 55–44% reduction for Delhi and Mumbai [65] |
| | Jiangsu, China | 19% reduction from pre-COVID; 23% increase post-COVID [75] |
| | Barcelona, Spain | 37% reduction [66] |
| | Valencia, Spain | 16.5% reduction [76] |
| | Makkah, Saudi Arabia | 12% reduction [70] |
| | Beijing-Tianjin-Hebei, China | 33.60% reduction [59] |
| | Leeds, Sheffield, and Manchester, England | 2.36–19.02% reduction [27] |
| | Delhi, India | 71% reduction [62] |

In particular, within the just-mentioned theoretical framework, studies show contradictory results of the above relation across different countries worldwide; these problems are a starting point for further investigation developed here. The study here endeavors to verify whether statistical evidence developed here supports the proposed *research hypothesis* that the general change (especially, reduction) of air pollutant concentrations between countries can be explained by the measures of control (restriction policy of lockdown) applied to face COVID-19 pandemic. The method of inquiry and results of the study here provide critical implications of strict public policies, beyond health emergencies, as measures of control for environmental pollution in order to foster sustainable pathways of growth. The next section presents the methods of inquiry for this purpose.

## 2. Materials and Methods

### 2.1. Research Setting and Sample

The current work focuses on countries of five geoeconomic regions: Asia, Europe, North America, South America, and Oceania, excluding Africa because there were not constantly available data for the examined periods. To explore air pollution changes nationally, we selected 19 countries and 300 cities around the world in order to obtain comprehensive data on air pollutants in the pre- and post-lockdown period to face the COVID-19 pandemic for appropriate statistical analyses.

The selected countries were:

- China, India, Israel, Japan, and South Korea from the Asian geoeconomic region;
- Croatia, Denmark, France, Germany, Macedonia, the Netherlands, Poland, Serbia, Spain, Turkey, and the United Kingdom from the European geoeconomic region;
- Canada and the United States from the North American geoeconomic region;
- Australia from the Oceanian geoeconomic region;
- Brazil, Chile and Colombia from the South American geoeconomic region.

### 2.2. Measures and Sources of Data

The analysis of the effects of the restriction policy of lockdown during the COVID-19 pandemic on air pollution is based on six fundamental air pollutants, given by:

$CO$, $NO_2$, $SO_2$, $O_3$, $PM_{2.5}$, and $PM_{10}$.

The measure of these air pollutants is based on the concentration of $\mu g/m^3$ (micrograms—one-millionth of a gram—per cubic meter of air).

Daily data and measures on six major air pollutants were obtained from the World Air Quality Index Portal from 2015 to 2022 [78].

### 2.3. Data Analysis Procedure

The average monthly concentrations of air pollutants from January to March in 2019, 2020, 2021, 2021 are compared to the same months of the 4-year baseline (2015–2018) to account for the impacts of containment policies of lockdown during COVID-19 on air pollution between countries.

The monthly relative rate of change (ROC) was preferred to make a comparison in the variance of air pollutants exposure in the different periods just mentioned. ROC is given by (Equation (1)):

$$ROC_m = \frac{[x2019m - x(2015 - 2018)m]}{x2019m} \times 100 \qquad (1)$$

where

$ROC_m$ = the monthly relative rate of change of pollutant concentration in month m
$x_{2019m}$ = the pollutant concentration in the month m over 2019 year
$x_{(2015–2018)m}$ = the baseline pollutant concentration in the month m from 2015 to 2018 period [79].

ROC values were calculated for January, February, and March from 2019 to 2022, keeping the baseline period between the years 2015 and 2018.

If $ROC_m$ was positive, the pollutant concentrations in 2019, 2020, 2021, and 2022 were higher than the reference period (2015–2018); instead, if $ROC_m$ was negative, the pollutant concentrations in 2019, 2020, 2021, and 2022 (during partial or full lockdowns) were lower than the baseline period, 2015–2018 [79].

## 3. Results and Discussion

The percentage variation in the concentration of air pollutants based on ROC during the lockdown periods compared with the baseline period (2015–2018) for 19 countries is presented in Table S1 in the Supplementary Materials. Although Figures 1–5 present mean concentration values in air pollutants between countries, the following discussion for countries in different geoeconomic regions is based on the index of monthly relative rate of change (ROC), calculated as described in the previous section.

### 3.1. Asia Geoeconomic Region

Variations in air pollutant concentrations in countries from the Asia geoeconomic region are visualized in Figure 1. When Figure 1 is examined for China, the air pollutant with the highest decrease in concentration values is $NO_2$ in January 2022 and January 2019, with a rate of 222% and 191%, followed by $SO_2$ in February 2021 and February 2020 with 171% and 161%, respectively (Figure 1, China). The times and rates of the maximum reduction in air pollutants other than $NO_2$ and $SO_2$ are CO with a rate of 137%, $O_3$ with a rate of 120%, $PM_{2.5}$ with a rate of 18% in March 2019, and $PM_{10}$ with a declining rate at 4% in February 2019. The full lockdown restrictions implemented in various cities in China, starting in January 2020, can be associated with these decreasing trends. Wu et al. [80] investigated how the lockdown to face the COVID-19 pandemic (from 1 January to 12 April 2020) in China affected traffic-based air pollutants in Shanghai, comparing the pollutant concentrations during the pandemic period with the data from 2018 and 2019. They observed a moderate decline in CO emissions, with a ratio of 28.8% and 16.4% for roadside and non-roadside stations, respectively. In South Korea, the lockdown measures for pandemic control started on 25 March 2020 [81]. $SO_2$ showed the highest decreasing trend in January 2022 (−113%) and January 2021 (−103%), compared with the baseline period 2015–2018. The rate of decrease in the concentrations of air pollutants varied between 2.3% and 113%. While ozone levels increased compared to the baseline period in all periods except March 2020, $PM_{2.5}$ increased in January of 2019, 2020, 2021, and 2022, and in February and March of 2019 (see Figure 1, South Korea). Vuong et al. [81] explored the influences of the quarantine measure of control on the difference in air pollutant concentrations in Daegu, South Korea. They observed reductions in the concentrations of air pollutants: a ratio of 3.75% ($PM_{10}$), 30.9% ($PM_{2.5}$), 36.7% ($NO_2$), 43.7% (CO), and 21.3% ($SO_2$). In India, $PM_{2.5}$ concentration values decreased in February of 2019 (−4%) and 2020 (−3.6%), while the remaining periods showed an increasing trend in which the maximum level was observed in March 2021 (28%). By contrast, $PM_{10}$ decreased in March 2019 (−13%) and 2020 (−18%), while there was an increase of 0.2% in January 2022 and 31.3% in March 2021. All air pollutants, except $PM_{2.5}$ and $PM_{10,}$ showed a decrease in the variation for almost

all investigated periods; the most effective decrease was observed for CO, with a rate of 147% in January 2022 when pollutant concentrations are compared with the average values of the 2015–2018 baseline period (see Figure 1, India). Declines in the concentrations of CO ($-84\%$), $NO_2$ ($-69\%$), $SO_2$ ($-5\%$), $O_3$ ($-32\%$), and $PM_{10}$ ($-18\%$) observed in March 2020 in India should be attributed to restrictions based on lockdown, which included the banning of all transport activities and the closure of industrial, commercial and private establishments, starting from March 2020. Singh and Chauhan [82] evaluated the impact of the total restriction of March 2020 on air quality parameters including $PM_{2.5}$, Air Quality Index (AQI), and tropospheric $NO_2$ in India using ground and satellite data. Results pointed out a declining trend in all air quality parameters studied (Figure 1, India).

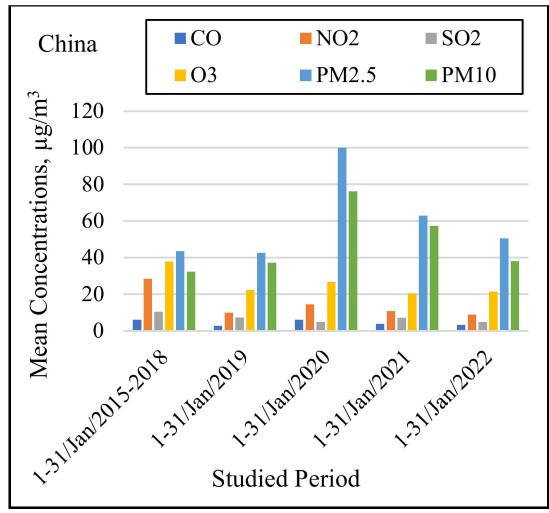
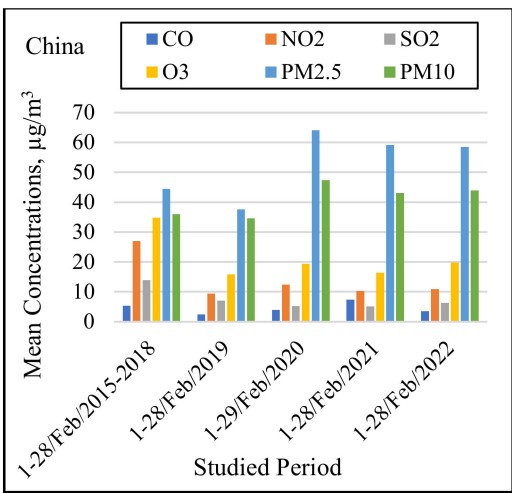
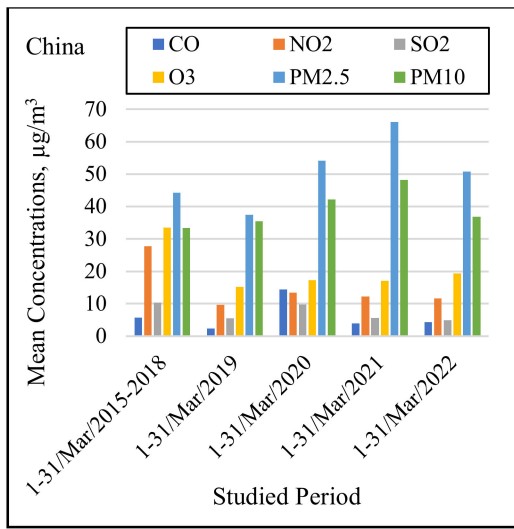
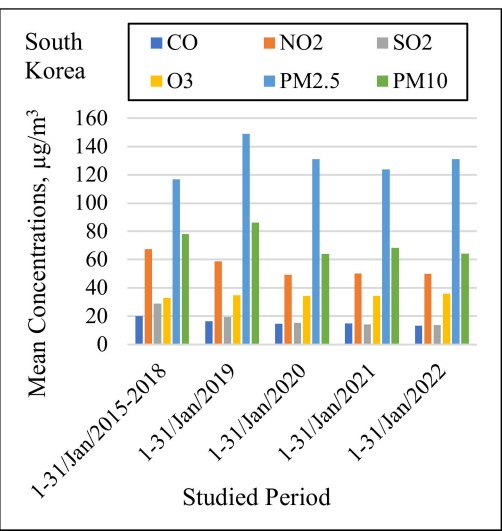

**Figure 1.** *Cont.*

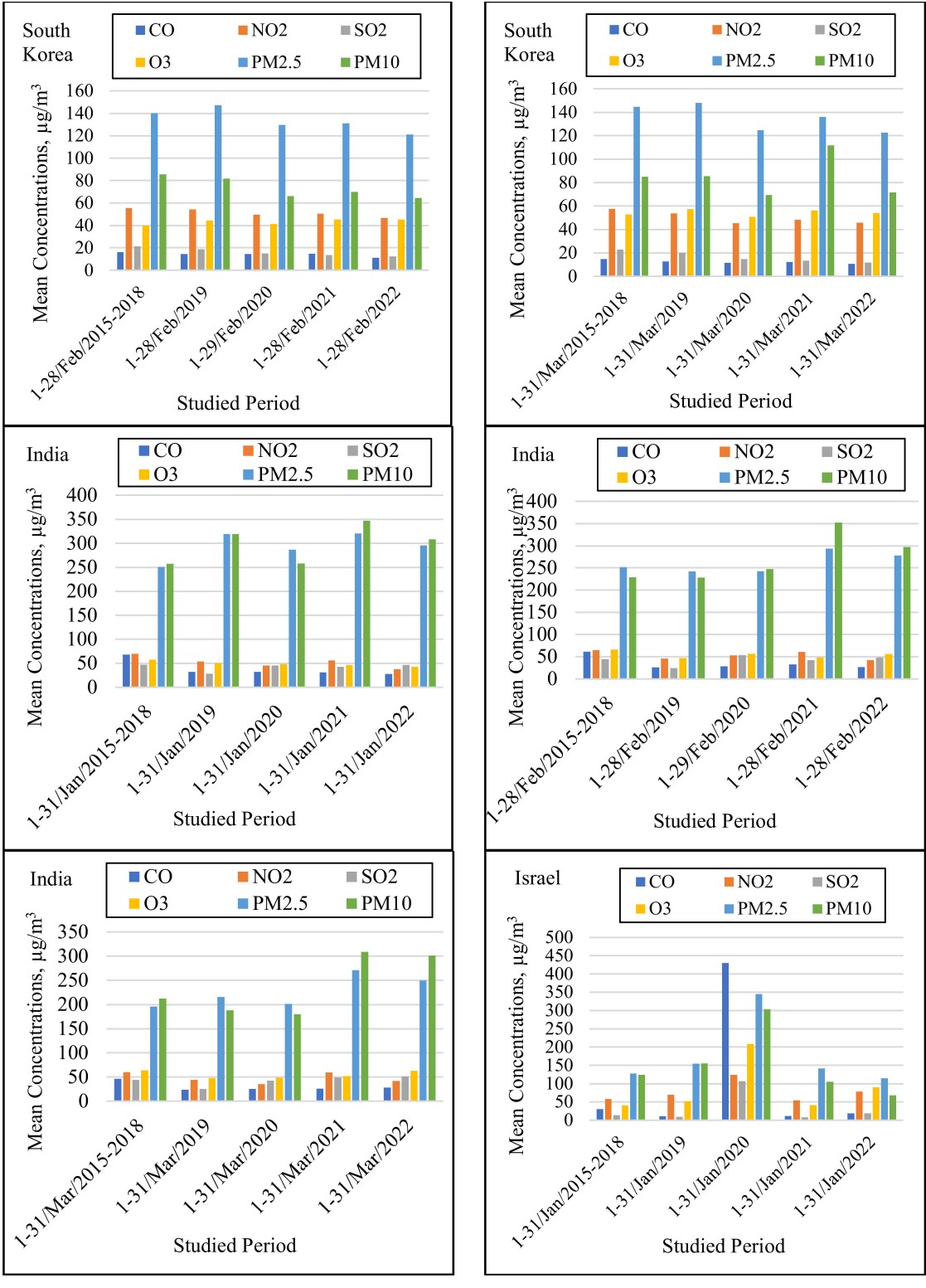

**Figure 1.** *Cont.*

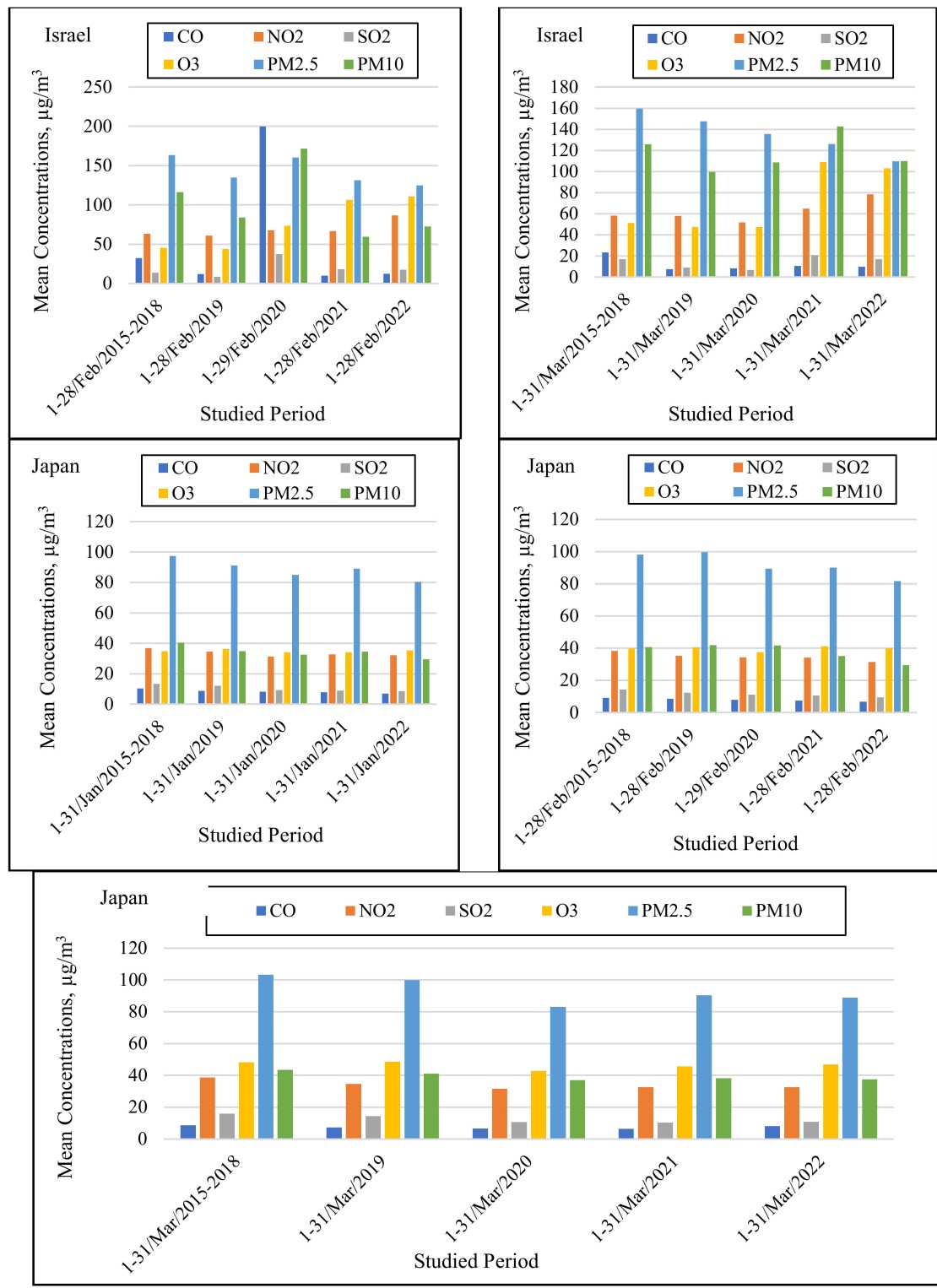

**Figure 1.** Average concentrations of air pollutant in countries of the Asian geoeconomic region.

In Israel, a remarkable decreasing trend was observed in CO (−180%) and SO$_2$ (−154%) in March 2020 which could be attributed to limitations imposed by Israel's government, including restrictions on the public and private sectors. The highest reduction in the pollutant concentrations for the studied period compared with the baseline period (2015–2018), known as the pre-COVID-19 period, was obtained in CO with a range from −59% to −226% (see Figure 1, Israel). Agami and Duyan [83] evaluated the impact of the COVID-

19 lockdown on air pollution in Haifa and Greater Tel Aviv, two regions with high air pollution in Israel. They found that pollution emissions were reduced during the COVID-19 lockdown relative to the same period in 2019. The biggest reduction was observed in $NO_x$, with an average value of 41%.

In Japan, the government announced a state of emergency on both 7 April 2020 and 8 January 2021 and applied soft measures of lockdown [84]. In January 2021, variations in the concentration of CO and $SO_2$ showed higher reductions of 34% and 12% in comparison with the same month of the previous year. Moreover, CO (−49%) and $SO_2$ (−56%) exhibited a maximum decrease in January 2022, $NO_2$ (−23%), $O_3$ (−13%), and $PM_{2.5}$ (−24%) in March 2020, and $PM_{10}$ showed the maximum reduction (−38%) in February 2022 compared with the baseline period (Figure 1, Japan). Hu et al. [85] explored the variation in levels of air pollution during and after the implementation of lockdowns in China, Japan, the Republic of Korea, and India, comparing the Air Quality Index (AQI) for the past three years. Results showed that the reduction in air pollutant levels during the examined periods between these cities was positively correlated. In Tokyo, low levels of air pollution were observed during the application of lockdown.

### 3.2. North America Region

Variations in air pollutant concentrations based on the pre- and post-COVID-19 period for the United States of America and Canada from the North American geoeconomic region are found in Figure 2.

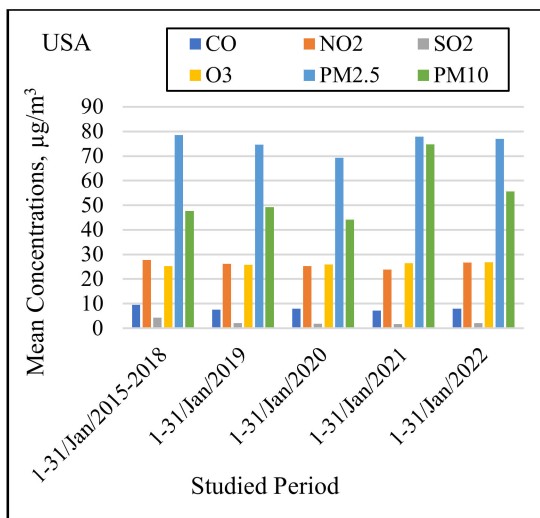
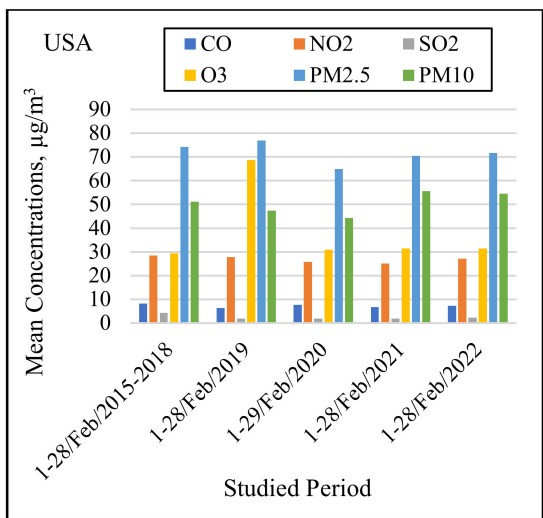

**Figure 2.** *Cont.*

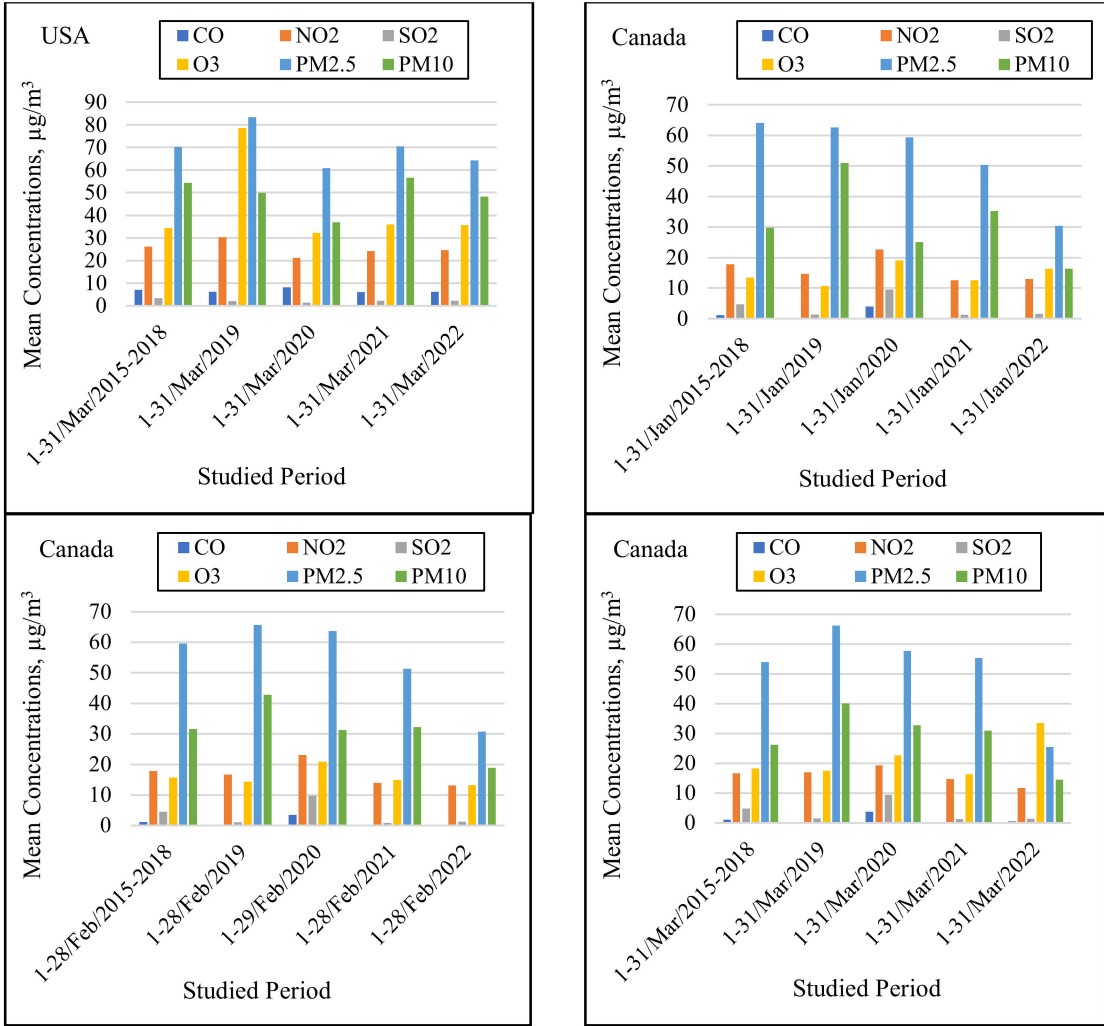

**Figure 2.** Average concentrations of air pollutant in countries of the North American geoeconomic region.

In the USA, a national emergency was announced in the second week of March 2020 with a closure of services and other activities [86]. In March 2020, reductions in the concentration of $NO_2$ (−23%), $SO_2$ (−137%), $O_3$ (−7%), $PM_{2.5}$ (−16%), and $PM_{10}$ (−47%) were observed. The highest decreasing ratio was obtained for $SO_2$ emissions compared with the pre-COVID-19 period (see Figure 2, USA). Shaakoor et al. (2020) investigated the changes in the air pollutants containing CO, $NO_2$, $SO_2$, $PM_{2.5}$ and $PM_{10}$ in the USA, considering the data during restriction periods of 2019 and 2020. The results showed that CO, $NO_2$, and $PM_{2.5}$ concentrations showed a decreasing trend of 19.3%, 36.7%, and 1.10%, respectively, while $PM_{10}$ and $SO_2$ increased by 27.8% and 3.81%, respectively, in five states of the USA during the period of lockdown.

In Canada, a national lockdown was declared from 22 March to 2 May 2020. Our results are found in Figure 2, Canada. Mashayekhi et al. [87] investigated the impact of the COVID-19 lockdown measures on air quality in Canada for the largest cities, including Toronto, Montreal, Vancouver, and Calgary, comparing the values of March–May 2020 with the same months of the previous 10 years (2010–2019). Results indicated that $NO_2$ and $PM_{2.5}$ demonstrated a decreasing trend in the presence of lockdown measures, while $O_3$ surface concentrations showed an increase up to a maximum of 21%.

*3.3. South America Region*

Results for countries of South America are in Figure 3.

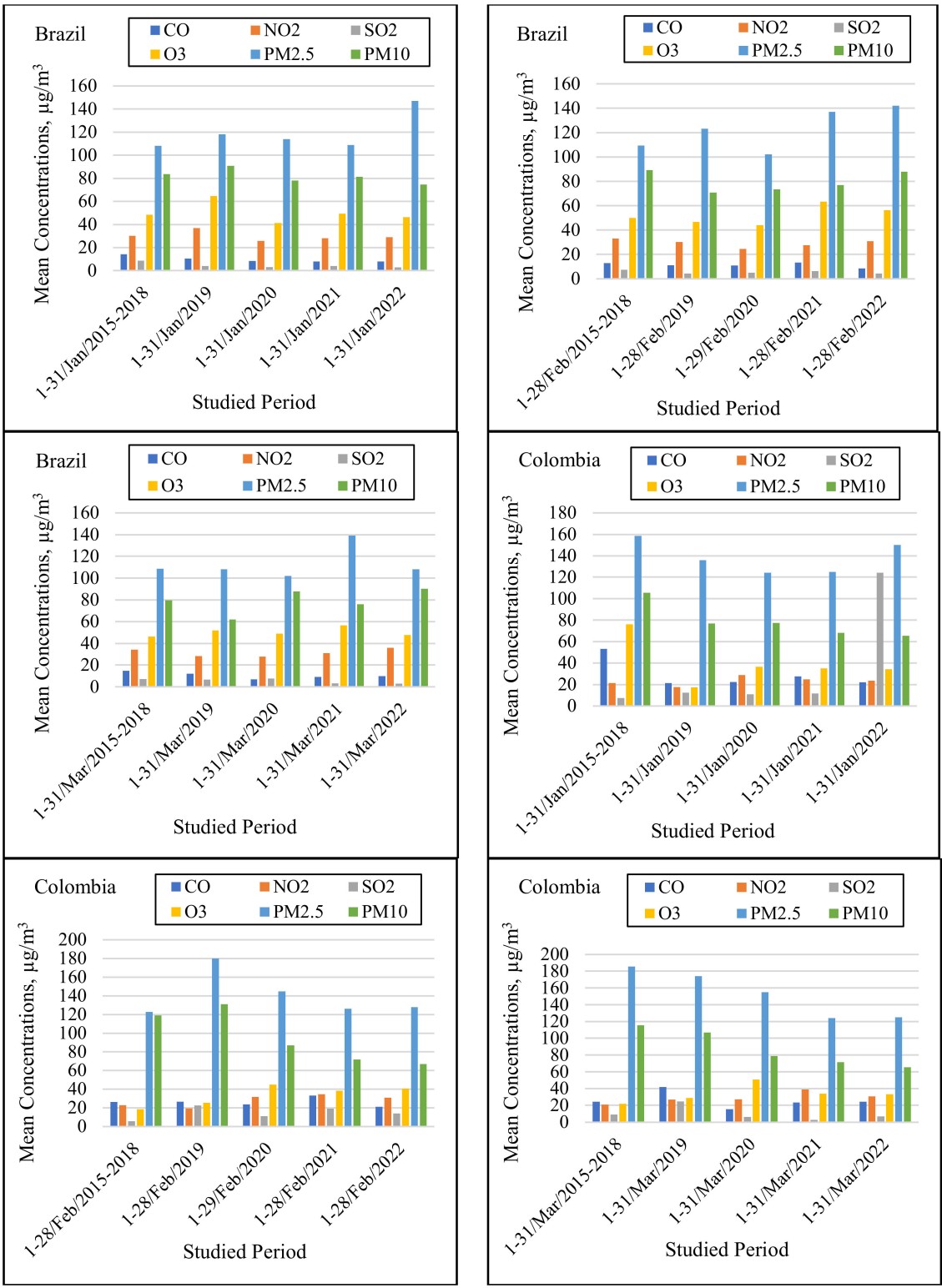

**Figure 3.** *Cont.*

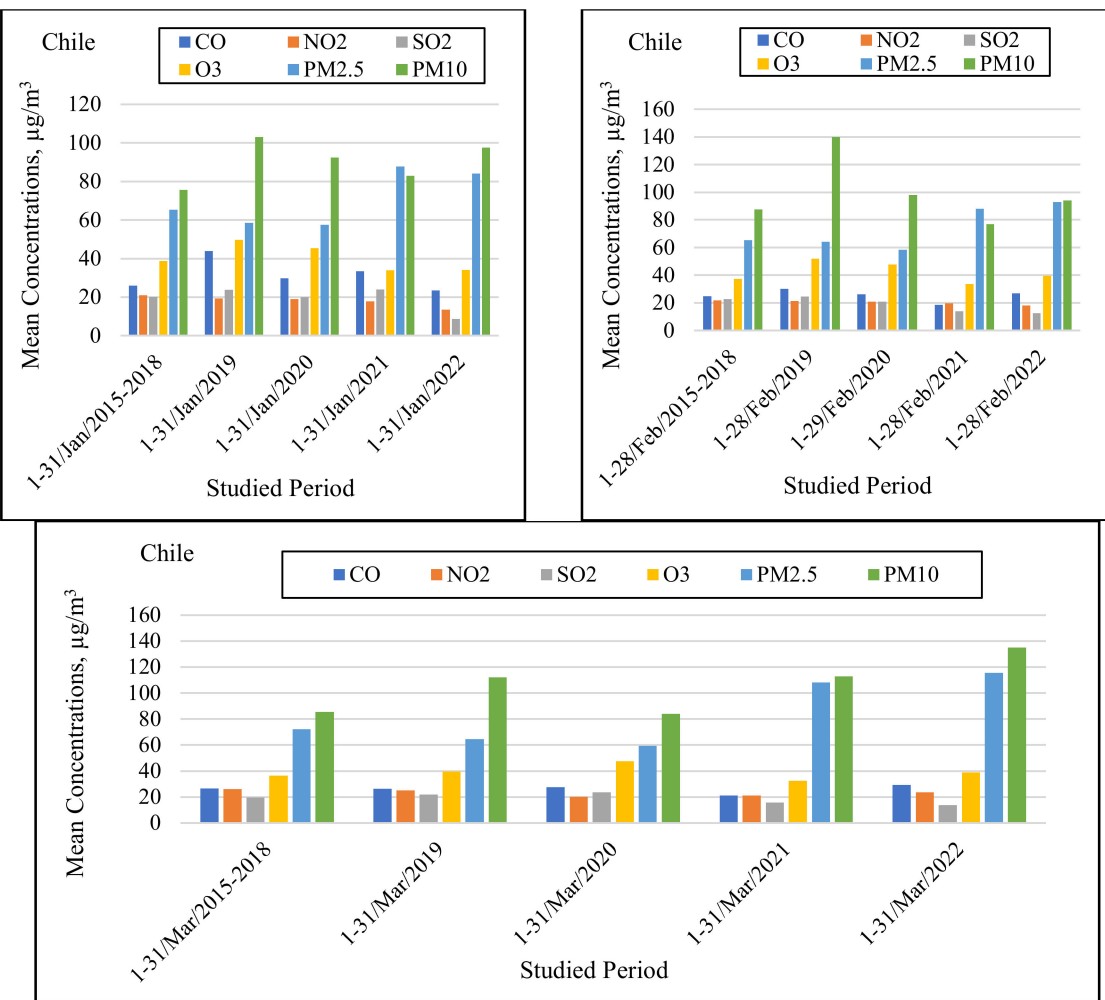

**Figure 3.** Average concentrations of air pollutant in countries of the South America.

In Brazil, a partial lockdown started in March 2020 and ended in June 2020 [88]. The influence of lockdown policies on the air quality of Brazil, considering the average air pollutant concentrations in March 2020 compared to the pre-COVID-19 period, showed that the highest decrease was from 22 to 112% for CO, followed by $NO_2$ (−24%, see Figure 3, Brazil). Beringui et al. [88], during the partial lockdown in Rio de Janeiro, Brazil, showed that CO concentration reduced significantly because of the decrease in the traffic density, while $O_3$ concentration increased, most probably due to a decline in primary air pollutants. In contrast, $PM_{10}$ concentration did not exhibit a remarkable variation. In short, the partial restrictions improved the air quality of Rio de Janeiro.

In Columbia, the government announced the quarantine restrictions on a national scale starting from 20 March to 31 August 2020. Variations of pollutant concentrations in March 2020, relative to the pre-COVID-19 period, had a reduction in CO (56%), $SO_2$ (39%), $PM_{2.5}$ (20%), and $PM_{10}$ (46%). Amaya and Samuel [57] compared the concentration levels of air pollutants in Bogota, Columbia, considering lockdown and previous periods. They observed a significant decline in pollutant concentrations by −13% and −22% in $NO_2$, −11% and −20% in $SO_2$, −23% and −34% in CO, −7% and −15% in $PM_{2.5}$, and −25% and −16% in $PM_{10}$, respectively. On the other hand, levels of atmospheric $O_3$ increased by 31% and 14% compared to the baseline period (Figure 3, Colombia). Finally, in Chile, lockdown restrictions started in March 2020; the highest variation of concentrations between the pre-COVID-19 period (2015–2018) and post-lockdown period was observed for $SO_2$ (−132%) in January 2022, whereas $NO_2$ (−29%), $PM_{2.5}$ (−21%), and $PM_{10}$ (−2%) showed decreasing trends in March 2020 (see Figure 3, Chile).

### 3.4. Europe

Figure 4 shows results for Europe. In Germany, a nationwide lockdown was imposed between 21 March and 30 June 2020 (Balamurugan et al., 2021). The maximum decreases in $NO_2$ (−30%) and $SO_2$ (−191%) concentrations were observed in January 2022, whereas the maximum decrease in $PM_{2.5}$ concentration was in February 2022 (Figure 4, Germany). In general, there is a decrease in the concentration of all parameters of air pollution associated with restrictions. Balamurugan et al. [18] found that human-based emissions in eight German metropolitan regions reduced by 23% for $NO_2$, while they increased by 4% for $O_3$ between the examined periods. In the Netherlands (Figure 4, The Netherlands), the restriction period based on lockdown was from 16 March 2020 to 10 May 2020 [89].The average values of the 2015–2018 baseline period were compared with values during the lockdown period; results showed a significant decrease in CO concentration in the Netherlands: $NO_2$ (−26%), $SO_2$ (−35%), $PM_{2.5}$ (−17%), and $PM_{10}$ (−18%) in March 2020 had lower levels than March in the 2015–2018 period, already before the lockdown period (see Figure 4, The Netherlands). Velders et al. [89] also investigated the Netherlands and showed that the lockdown reduced observed $NO_2$ concentrations with larger values than results obtained from simulation models.

In Spain, the lockdown was applied from March 2020 to June 2020 [34]. CO showed the maximum decrease in its concentration during the examined months in 2022. Comparative analysis with the baseline period also showed reductions in $NO_2$ (−57%), $SO_2$ (−46%), and $PM_{10}$ (−15%) (see Figure 4, Spain). Donzelli et al. [34] compared the air pollution level in Valencia, Spain between lockdown and non-lockdown periods. The highest reductions in the $PM_{10}$ and $PM_{2.5}$ levels were in the range of 56–60% and 41–53%.

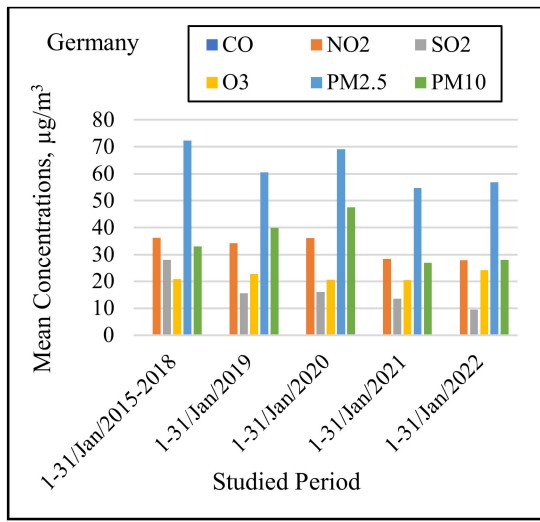 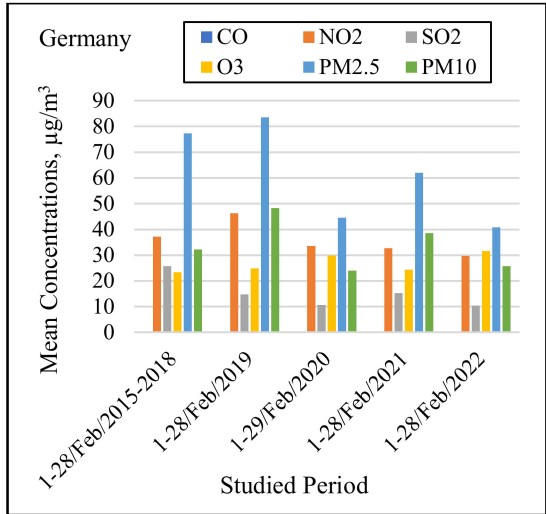

**Figure 4.** *Cont.*

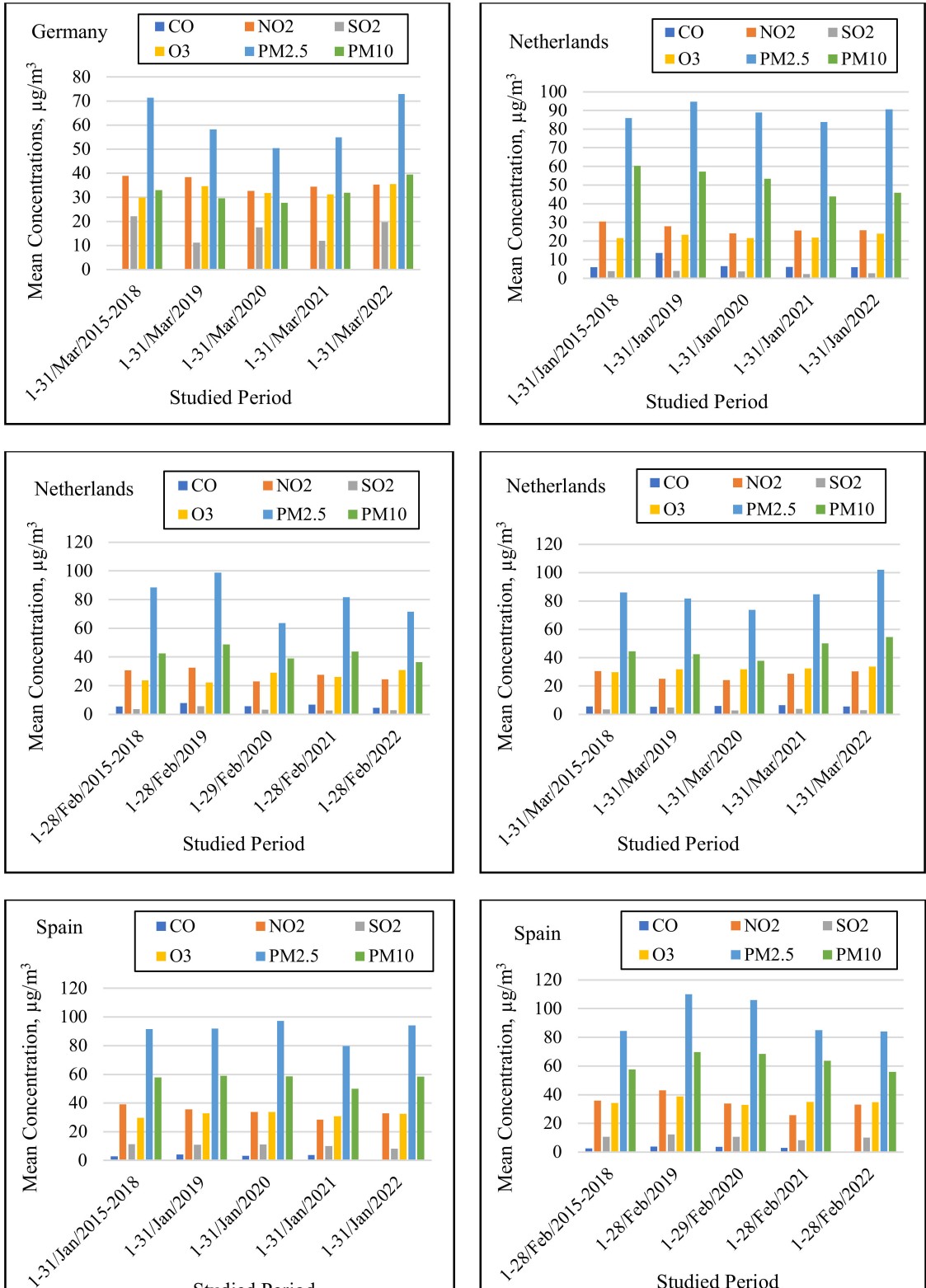

**Figure 4.** *Cont.*

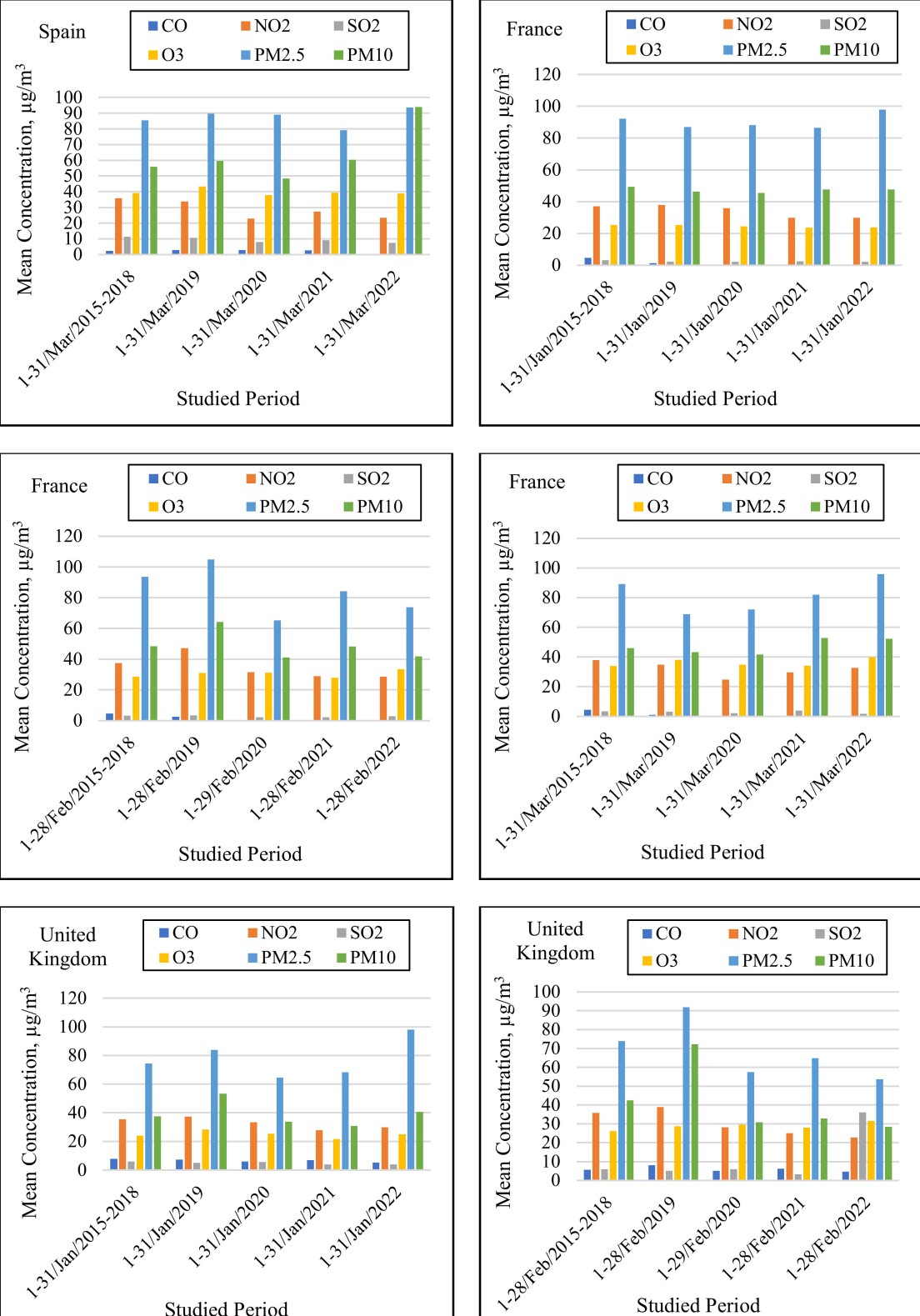

**Figure 4.** *Cont.*

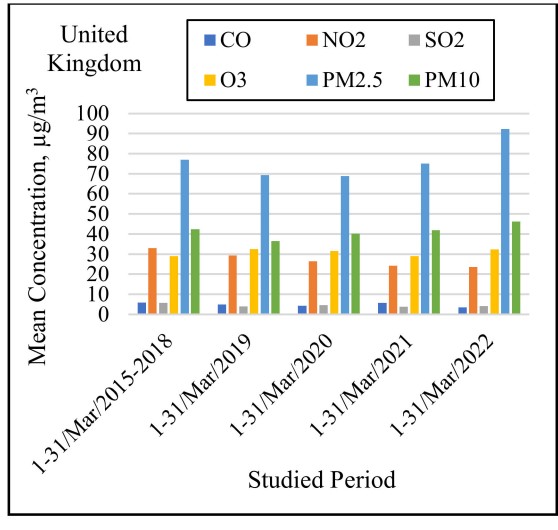

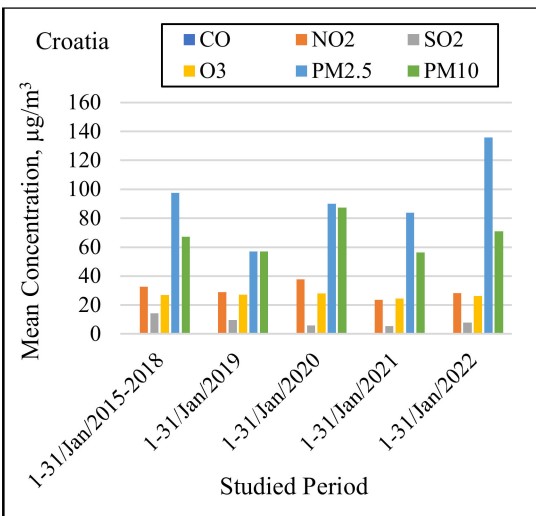

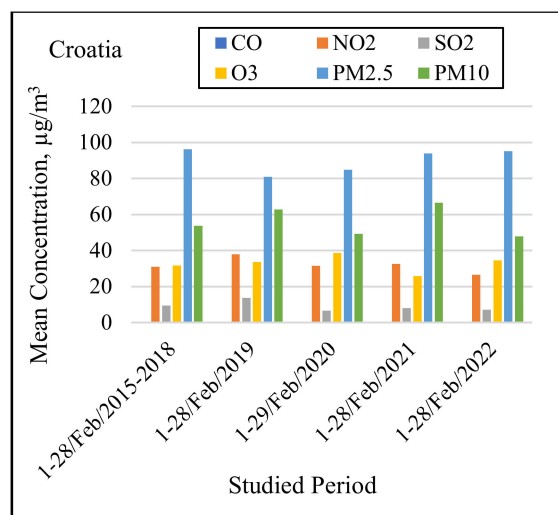

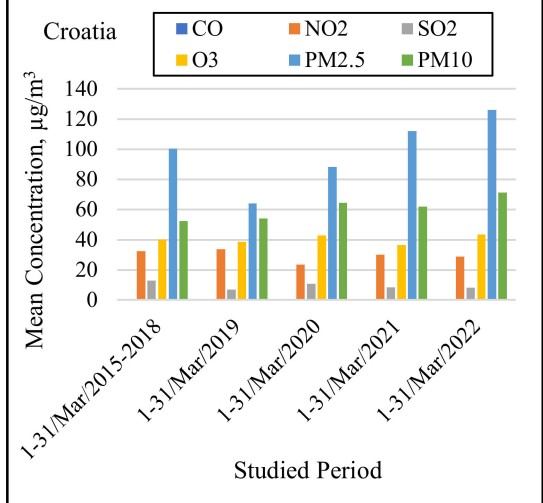

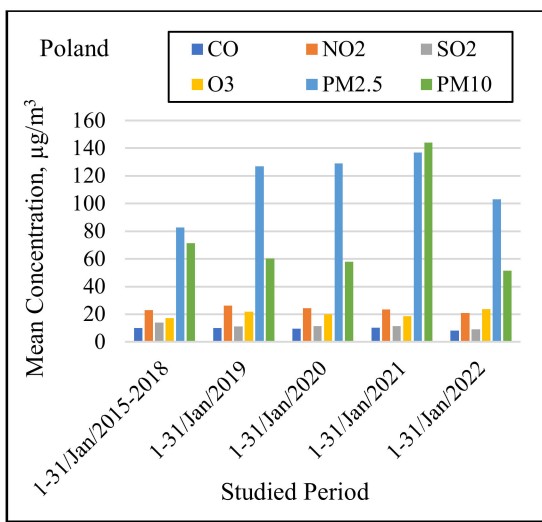

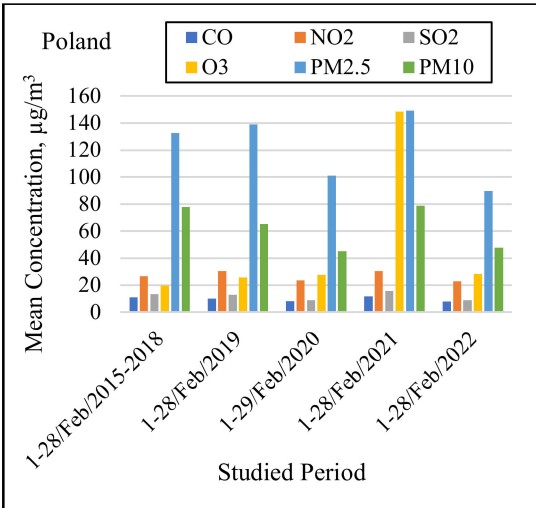

**Figure 4.** *Cont.*

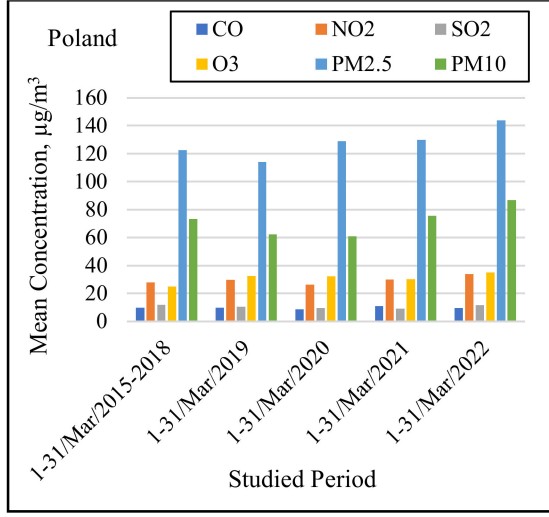
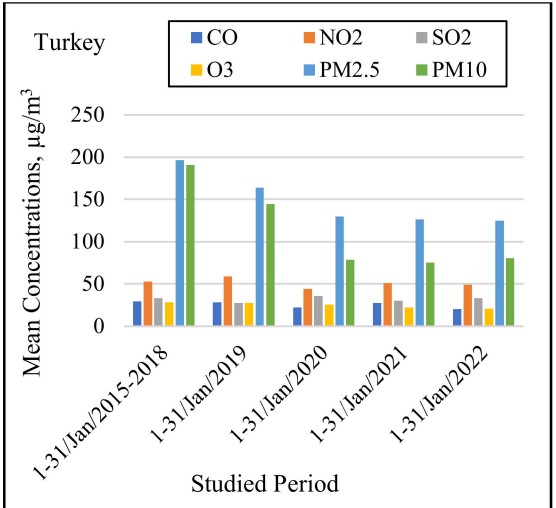
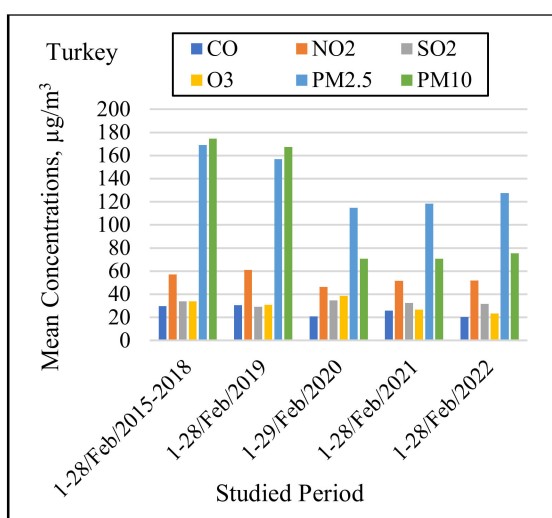
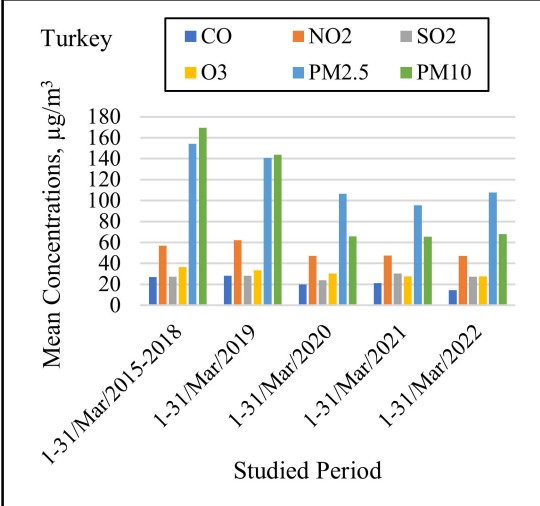

**Figure 4.** Average concentrations of air pollutant in countries of the European geoeconomic region. Note: In Germany, the concentration of CO varied in the range of 0.10–0.26 $\mu g/m^3$ for the examined periods. Since CO concentration was low compared with the concentrations of other pollutants, it is not included in the graphs.

In France, the lockdown restrictions were from 17 March 2020 to 11 May 2020 [90]. An extremely remarkable decrease in CO concentrations, up to over 4500%, was observed for almost all periods under study. In general, all pollutants decreased compared to the baseline period: $NO_2$ (−53%), $SO_2$ (−62%), $PM_{2.5}$ (−24%), and $PM_{10}$ (−10%) concentrations (see Figure 4, France). Ikhlasse et al. [90] also indicated that the maximum daily concentrations of the air quality parameters showed decreasing levels in the range of 5.1 and 44.48%.

In the United Kingdom, the lockdown was announced starting on 24 March 2020 because of the rapid increase in confirmed cases [71]. Pollutants mostly decreased compared to the baseline period: March 2022 for CO (−71%), February 2022 for $NO_2$ (−57%), February 2021 for $SO_2$ (−79.5), January 2021 for $O_3$ (−11%), February 2022 for $PM_{2.5}$ (−38%) and February 2022 for $PM_{10}$ (−49%). In addition, in March 2020, when the restriction policies started, all concentrations of air pollutants except ozone showed decreasing trends (see Figure 4, The United Kingdom). Jephcote et al. [71], investigated the changes in air pollutant concentrations in the United Kingdom by comparing daily pollutant measurements of $NO_2$, $O_3$, and $PM_{2.5}$ during the lockdown period (from 30 March 2020 to 3 May 2020) with the 2017–2019 period. Results revealed that $NO_2$ and $PM_{2.5}$ concentrations reduced, while $O_3$ concentrations increased.

In Croatia, the lockdown restrictions started on 16 March 2020 [91]. During these restrictions, $NO_2$, $SO_2$, and $PM_{2.5}$ concentrations exhibited reductions compared with average values of the previous period, given as $-37.9\%$, $-18.6\%$, and $-14\%$, respectively. $SO_2$ exhibited the highest reduction, in January 2021 ($-169\%$), while CO concentration did not change (see Figure 4, Croatia). Jakovljević et al. [92] compared concentrations of $PM_1$ and $NO_2$ between the lockdown period and the previous year's period and found a remarkable decrease in pollutant concentrations associated with traffic emissions.

In Poland, the lockdown policy was imposed on 12 March 2020 [35]. The maximum reduction for CO ($-41\%$), $NO_2$ ($-16\%$), $PM_{2.5}$ ($-48\%$), and $PM_{10}$ ($-63\%$) was recorded in February 2022, while the highest decrease in $SO_2$ was observed in January 2022 (Figure 4, Poland). Reductions were also observed in pollutant concentrations (CO, $NO_2$, $SO_2$, $PM_{10}$) in March 2020, when containment policies started. Filonchyk et al. [35] measured atmospheric pollutants in Poland during the policies of restrictions and found significant reductions compared with values in 2018 and 2019.

Finally, in Turkey, the government announced the first restrictions on March 12, 2020 [93]. In March 2020, variations in the concentrations of all air pollutants were: $-36\%$ for CO, $-21\%$ for $NO_2$, $-14\%$ for $SO_2$, $-20\%$ for $O_3$, $-45\%$ for $PM_{2.5}$, and $-157\%$ for $PM_{10}$, respectively (compared with the average values of the pre-pandemic period 2015–2018). Among these air pollutants, $SO_2$, $O_3$, and $PM_{10}$ showed a tendency to decrease more in March 2020, also without lockdown measures. Except for February 2019, maximum reductions were recorded for $PM_{10}$ pollutant concentration for all periods investigated (see Figure 4, Turkey). Orak and Ozdemir [93] compared air pollutant concentrations in Turkey during lockdown restrictions with the previous five years' data. Results suggested that $PM_{10}$ and $SO_2$ concentrations are due to transportation emissions based on workplace mobility.

*3.5. Oceania*

As far as the Oceania geoeconomic region goes, changes in air pollutant concentrations are in Figure 5. In Australia, the lockdown started on 16 March 2020 [94]. All air pollutant concentrations, except $PM_{2.5}$ and $PM_{10}$, decreased in March 2020, and the maximum reductions were by $NO_2$ at 106%, $O_3$ at 94%, and CO at 61%. When all investigated periods were compared with the average values of the pre-COVID-19 pandemic period, $NO_2$ showed the maximum decrease except for January–February 2020 and February 2021 (see Figure 5, Australia). Duc et al. [94] evaluated the lockdown impact in Australia on the air pollutant concentrations, and they found that all pollutants examined except for ozone revealed a drop in their concentrations during the period having restriction policies.

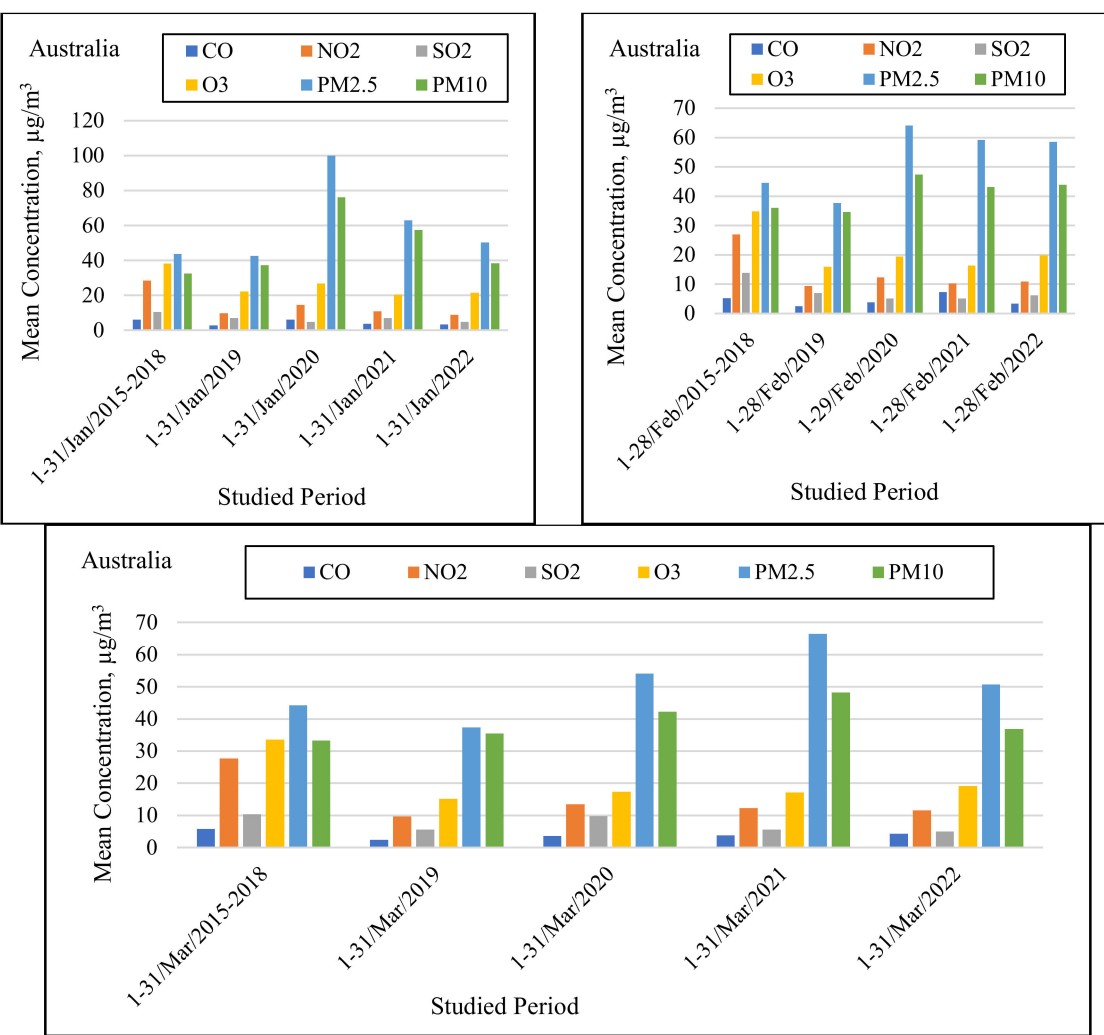

**Figure 5.** Average concentrations of air pollutant in a country of the Oceania geoeconomic region.

*3.6. General Observations*

Considering the restriction policies imposed in manifold countries under study, when the pollutant concentration is compared with the pre-COVID-19 period, the maximum CO reduction was in France with a rate of 4668% in January 2020, 2021, and 2022, followed by Canada with 1010% in January 2019, 2021 and 2022. Comparative analysis also showed that Australia and China have a maximum and equal decrease of 222% in January 2022 [95]. The highest declining rate of $SO_2$ (generated by both natural sources, such as volcanoes, and anthropogenic sources, such as coal-burning power plants, smelters, and oil refineries) is for Canada (413%) in February 2021, Colombia (224%) in March 2021, and Germany (191%) in January 2022 [96]. The study here also shows that the country with the maximum decrease (−343%) in ozone concentration at ground level (comparing values during the COVID-19 pandemic with lockdown and the data of the 2015–2018 period) is Colombia, which also showed a maximum decrease in $NO_2$ (−21%) and CO (−151%) concentrations. The countries with maximum declines in PM emissions are Canada, with a decrease of 112% for $PM_{2.5}$ over March 2022, and Turkey, with a decrease of 159% for $PM_{10}$ over March 2021.

Table 2 summarizes the main effects of the lockdown policy to face COVID-19 on air pollution between different countries. The maximum reduction in CO emissions was recorded in India, Israel, Canada, France, Germany, and Spain; the minimum was in China, Poland, and Australia. Moreover, the maximum reduction in $NO_2$ emissions was observed only in China and Australia, whereas the minimum decrease was observed in Chile. Maximum reductions for $SO_2$ emissions were recorded in Japan, South Korea, America, Brazil,

Croatia, Netherlands, Poland, and the United Kingdom, while minimum reductions were recorded in Croatia, Spain, and Turkey. In addition, no country recorded a maximum reduction in $O_3$ and $PM_{2.5}$ concentrations. The minimum reduction in $O_3$ concentrations was observed in Japan and Croatia, while Israel, South Korea, the Netherlands, and England were the countries where the minimum reduction was observed in $PM_{2.5}$ among air pollutants. As for $PM_{10}$, Colombia and Turkey were the countries representing the maximum decrease in its concentrations; by contrast, Canada, Brazil, America, France, and Germany are the countries showing the highest decrease in $PM_{10}$ concentrations. Overall, when the maximum reduction in pollutant concentrations is evaluated in terms of amount, then CO, with a rate of 655.5%, is superior to others, whereas, if the sum of the countries is evaluated, $SO_2$ shows a decrease in nine different countries.

**Table 2.** Main effects of the containment policy of lockdown to face COVID-19 on average levels of air pollutants between selected countries worldwide.

| Geoeconomic Region | Country | Average Variation % of Air Pollutant Values from January 2019 to March 2022 Compared with Baseline Period (2015–2018) | | | | | |
|---|---|---|---|---|---|---|---|
| | | CO | NO₂ | SO₂ | O₃ | PM₂.₅ | PM₁₀ |
| Asia | China | −49.68 | −153.77 | −98.25 | −87.26 | 17.10 | 20.89 |
| | India | −105.90 | −39.30 | −17.20 | −24.50 | 12.40 | 13.10 |
| | Israel | −122.10 | 12.30 | −13.80 | 33.30 | −8.20 | −18.80 |
| | Japan | −24.50 | −15.00 | −37.50 | −1.80 | −12.30 | −16.10 |
| | South Korea | −26.70 | −20.40 | −64.70 | 5.90 | −1.30 | −12.90 |
| North America | Canada | −655.50 | −13.80 | −193.70 | 4.50 | −27.30 | −8.10 |
| | USA | −17.10 | −7.60 | −107.30 | 12.10 | −4.30 | −2.30 |
| South America | Brazil | −48.50 | −10.90 | −89.00 | 4.70 | 8.60 | −7.70 |
| | Chile | 4.30 | −15.80 | −24.40 | 6.70 | 8.60 | 16.70 |
| | Colombia | −46.10 | 19.00 | 18.50 | −29.30 | −12.40 | −46.30 |
| Europe | Croatia | 0.00 | −8.10 | −60.10 | −0.02 | −11.90 | 5.70 |
| | France | −3471.80 | −17.90 | −33.50 | 3.60 | −11.40 | −1.80 |
| | Germany | −135.00 | −11.50 | −92.20 | 9.80 | −29.60 | −1.60 |
| | Netherlands | 10.50 | −16.00 | −16.30 | 8.20 | −4.50 | −7.70 |
| | Poland | −9.60 | 2.90 | −24.30 | 27.30 | 7.00 | −16.80 |
| | Spain | −73.80 | −22.20 | −18.50 | 4.20 | 3.90 | 5.50 |
| | Turkey | −27.60 | −9.40 | −4.40 | −20.40 | −39.50 | −115.00 |
| | United Kingdom | −19.40 | −23.40 | −26.20 | 7.20 | −4.90 | −7.00 |
| Oceania | Australia | −59.80 | −153.70 | −98.30 | −87.30 | 17.10 | 20.90 |

Note: For details see Supplementary Materials.

Chossière et al. [97] investigated the impact of quarantine measures on air pollution levels, and they found that $NO_2$ concentrations displayed a reduction on a global scale, while $PM_{2.5}$ and $O_3$ did not show any reduction in their concentrations based on the restrictions. Dang and Trinh [98] investigated air quality levels globally, and they indicated that lockdown had a positive impact on air quality by decreasing $NO_2$ and $PM_{2.5}$ concentrations. Hammer et al. [99] compared global $PM_{2.5}$ concentrations in 2020 based on lockdown measures of pandemic control with the concentrations in 2018 and 2019 for different geoeconomic regions: in general, $PM_{2.5}$ concentrations declined in the examined countries.

He et al. [79] investigated variations in the concentration of air pollutants globally, comparing data from the lockdown period in 2020 and baseline period (2015–2019). They found that concentrations of $PM_{2.5}$ and $NO_2$ decreased, whereas $O_3$ concentration increased. Kumari and Toshniwal [65] analyzed the variations in the concentration of air pollutants between the year with restriction measures and pre-lockdown periods. They observed that the concentrations of $PM_{2.5}$, $PM_{10}$, and $NO_2$ significantly decreased, whereas $SO_2$ slightly

reduced owing to the contribution of power plants. Torkmahalleh et al. [100] investigated the global changes in the concentration of air pollutants because of lockdown measures. The results confirm other studies indicating a decrease in the $NO_2$ and $PM_{2.5}$ concentrations, and an increase in the $O_3$ concentration.

Hence, the measures of control applied by countries to contain the negative impact of the COVID-19 pandemic can reduce the sources of air pollutants that contribute to release greenhouse gas emissions into the atmosphere. In particular, since the major sources of $SO_2$ and CO are the combustion of fossil fuels containing sulfur and carbon, particularly from power stations burning coal and vehicles [101], and for $NO_2$ and tropospheric $O_3$ the major sources are vehicle emissions and industrial processes [102], the restriction policies of lockdown to face the COVID-19 pandemic can reduce the pollutant concentrations mainly associated with the consequential limitations in transportation.

### 4. Concluding Remarks

In general, the results of the study here, using updated data, are consistent with previous studies and lead to the main findings that containment strategies to prevent the rapid diffusion of COVID-19 pandemic (e.g., lockdown) also reduced air pollution, improving temporary air quality and the environment with a positive societal impact for the health of people.

In particular, the statistical evidence above seems in general to support the *research hypothesis* stated in the Introduction that the rate of change (reduction) of air pollutants can be explained by the restriction policies of lockdown applied by countries to face the rapid diffusion of the COVID-19 pandemic.

Moreover, the findings here revealed that the effects of national restrictions on air quality vary significantly between countries. In fact, in March 2020, the countries that showed the maximum reduction in pollutant concentrations are France with CO ($-4367.5\%$), China and Australia for $NO_2$ ($-150.5\%$), Israel for $SO_2$ ($-154.1\%$), China for $O_3$ ($-94.1\%$), Germany for $PM_{2.5}$ ($-41.4\%$) and Turkey for $PM_{10}$ ($-157.4\%$). The variability of the observed improvements in air quality between regions may be due to the different geoeconomic, socioeconomic, environmental, climate, and atmospheric characteristics. These results suggest that the COVID-19 control measures imposed by governments of countries brought about a substantial decline in the concentration of air pollutants in contrast to pre-lockdown periods. It has been observed that the countries showed a greater decrease in air pollutant concentrations in March 2020, when they imposed mostly full lockdowns that led to constraints of the pollutant sources, such as transportation based on burning of gasoline and diesel, and industrial activities releasing a significant amount of greenhouse gases into the atmosphere. Manifold studies confirm that various strict measures of control taken by governments can mitigate/stop the spread of COVID-19 and also affect levels of air pollution in regions, comparing the primary and secondary air pollutant concentration values from air quality monitoring stations of COVID-19 pandemic period with the data of previous years. The majority of the results suggests that the concentrations of air pollutants substantially decreased during the period of lockdown, while the ozone concentration generally increased because of the decrease in nitrogen dioxide emissions, especially from motor vehicles and industrial activities. This study shows consistent results with previous literature and extends knowledge of the critical role of important and drastic interventions of public policy to face health emergencies, reduce air pollutants, and improve air quality and the environment in countries. However, it is also important to observe that measures of control to decrease the rapid spread of SARS-CoV-2 and cope with the COVID-19 pandemic crisis, such as lockdown, have pros and cons: they reduce, whenever possible, transmission dynamics and air pollution, but they have also negative consequences on economic systems, including integrated solid waste management (Figure 6). In short, strict restriction measures imposed by governments to combat the transmission of the COVID-19 pandemic (tracing systems, lockdown and quarantine, etc.) can improve air quality by mitigating greenhouse gas emissions and air pollutants based on anthropogenic sources, but they

also can increase socioeconomic issues and mental health problems and the consequential diffusion in the environment of micropollutants in wastewater from the consumption of antidepressants, antibiotics and other drugs, as well as the increase of waste caused particularly by medicine usage, etc. [23,103–109].

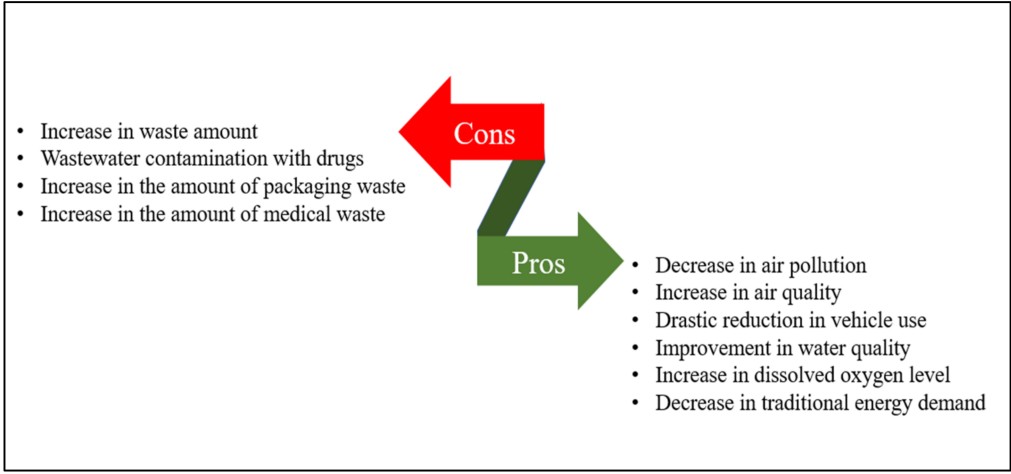

**Figure 6.** Pros and cons of COVID-19 lockdown restrictions on the environment.

Even though this study has presented interesting findings, based on a new dataset, which are of course tentative, it has also limitations. First, seasonal variations in pollutant concentration are not comprehensive because of the unavailability of data for many countries. Second, not all confounding factors that influence air pollution are considered during the measures of control implemented by countries; a study by Jakob et al. [46] stated that the decrease in pollutant concentrations observed may not be related to the restrictive measures for COVID-19 but it can be associated with seasonal changes. In future developments of this study, for confirming the results proposed here, these aspects have to be evaluated. In addition, the structure of pollution sources should also be analyzed and, considering the variability between countries, there needs to be specific case studies per country based on different types of industries (manufacturing, transport, energy, etc.). Finally, the extension of the period and countries under study with the update of data is required to reinforce the results of statistical analyses here and to truly warrant that these findings can support effective policies of crisis management for the next pandemic, considering health and environmental aspects.

Overall, then, measures of control for COVID-19 affect air quality and the general environment and they should be designed considering manifold aspects including economic and social factors. In this context, considering the expectations that containment measures will play a critical role in determining future policy actions to fight against pandemic threats, future studies can analyze the general effects of containment policies on the environmental pollution, also examining how air pollutant concentrations change seasonally in the long term.

To conclude, socioeconomic and environmental factors should be included in a general public policy of containment based on good governance, high investments in the health sector, and science advances to cope with new infectious diseases effectively and to support fruitful pathways of sustainable development, having positive social effects for the wellbeing and health of people [5,9,104,106,108].

**Supplementary Materials:** The following supporting information can be downloaded at: https://www.mdpi.com/article/10.3390/app122412806/s1, Table S1: Variation % in the concentration of air pollutants during the lockdown periods compared with baseline period (2015–2018) for 19 countries worldwide.

**Author Contributions:** A.P.A. and M.C.: Conceptualization, A.P.A. and M.C.; methodology, A.P.A.; software, A.P.A.; validation, A.P.A. and M.C.; formal analysis, A.P.A.; investigation, A.P.A.; resources, A.P.A. and M.C.; data curation, A.P.A. and M.C.; writing—original draft preparation, A.P.A.; writing—review and editing, A.P.A. and M.C.; visualization, A.P.A. and M.C.; supervision, M.C. All authors have read and agreed to the published version of the manuscript.

**Funding:** This study has no specific grant from any funding agency either official or commercial.

**Institutional Review Board Statement:** Not applicable.

**Informed Consent Statement:** Not applicable.

**Data Availability Statement:** The datasets used during the current study are available from the author on reasonable request.

**Conflicts of Interest:** The authors declare no conflict of interest.

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
