# Peer review of "Changes of Air Pollution between Countries Because of Lockdowns to Face COVID-19 Pandemic"

_applsci, doi:10.3390/app122412806_

Round 1

Reviewer 1 Report

The authors extensively reviewed previous similar studies. For their own analysis, the authors used air quality data from the World Air Quality Index Portal. The article corresponds to the topic of the journal, but minor revisions should be made according to the comments provided:

* The end of the introduction lacks a clearly formulated main purpose of the article. 

* Results and Discussions chapter. The main shortcoming of the article is related to the analysis of the results. It is not enough to describe how much the concentration decreases or increases. The structure of pollution sources must also be evaluated (various types of industry, transport, energy, households, etc.). It will be possible to answer not only how much the concentration decreased/increased, but also to analyze why it happened.

* In Figures 1-5, the results of individual months (January, February, March) should be more clearly distinguished.

* There are no CO concentrations among the Germany results (Fig. 4). They need to be added or corrected in the legend of this figure.

* Fig. 6. needs to be adjusted. "Economic losses" do not belong to the environment. I don't understand why "reduction waste recycling" appears. Instead of "waste content" you should write "waste amount". Instead of "contamination of land, water and air" it should be written more precisely: "wastewater contamination with drugs". In addition, "increase in the amount of packaging waste" and "increase in the amount of medical waste" should be mentioned.

* Conclusions must be formulated according to the authors' own results. You don't need to rely so much on other authors (with references). This is not a discussion chapter, but a conclusion chapter.

Author Response

The authors extensively reviewed previous similar studies. For their own analysis, the authors used air quality data from the World Air Quality Index Portal. The article corresponds to the topic of the journal, but minor revisions should be made according to the comments provided:
* The end of the introduction lacks a clearly formulated main purpose of the article. 
--Authors’ comments:
Thanks for this comment. Now the introduction has clarified the goal as follows:
“The principal goal of the paper here is to expand these studies in order to clarify the relationship between containment policy of lockdown and levels of air pollution using new data until August 2022 between different countries worldwide. In particular, this study here analyzes how levels of air pollutants for CO, NO2, SO2, O3, PM2.5, and PM10 change with restriction policy (lockdown) between nineteen countries from five geoeconomic regions from 2015 to 2022. 
In particular,  within  just mentioned theoretical framework, the limited evidence worldwide cannot satisfactorily explain the above relation and this is a starting point for further investigation developed here. These problems that the study here endeavors  to explain, are the foundations to see whether statistical evidence that is developed here supports the proposed research hypothesis that the general  change (especially, reduction) of air pollutant concentrations between countries can be explained by measure of control (restriction policy of lockdown) applied to face COVID-19 pandemic. The method of inquiry and results of the study here have critical implication on the role of strict public policies, beyond health emergencies,  as measures of control of environmental pollution to foster sustainable pathways of growth. The next section presents the methods of inquiry for this purpose.”

* Results and Discussions chapter. The main shortcoming of the article is related to the analysis of the results. It is not enough to describe how much the concentration decreases or increases. The structure of pollution sources must also be evaluated (various types of industry, transport, energy, households, etc.). It will be possible to answer not only how much the concentration decreased/increased, but also to analyze why it happened.
--Authors’ comments:
Thanks for this comment. Now the text has clarified these aspects.
We added some references indicating the sources of the pollutants and explaining, whenever possible, pollution sources, though there is a variability between countries that deserve specific case studies. We added this explanation at General Observation part as below: 
“Considering the measures applied by the countries to prevent the negative outbreaks of the COVID-19 outbreak due to its airborne transmission indirectly, remarkable drops can be expected in the sources of air pollutants, such as burning fuels for road transport and electricity generation, which contribute to release greenhouse gas emissions into the atmosphere. It is well-known fact that the major sources of SO2 and CO are the combustion of fossil fuels containing sulfur and carbon, particularly from power stations burning coal and vehicles (Holman, 1999). As for NO2 and tropospheric O3, vehicle emissions and industrial processes are the major contributors (Ciencewicki and Jaspers, 2007). In light of this information, the reductions observed in pollutant concentrations during quarantine periods based on the strict restrictions applied by countries can be attributed to the limitations in especially transportation”.

* In Figures 1-5, the results of individual months (January, February, March) should be more clearly distinguished.

--Authors’ comments:
Thanks for this comment. Figures have been revised accordingly.

* There are no CO concentrations among the Germany results (Fig. 4). They need to be added or corrected in the legend of this figure.
--Authors’ comments:
Thanks for this comment. 
We  have added a note to Figure 4 explaining why there are no CO concentration results for Germany as below: 
“Note: In Germany, the concentration of CO varied in the range of 0.10 – 0.26 µg/m3 for the examined periods. Since CO concentrations were so low compared with the concentrations of other pollutants, they were not in the graphs”.

* Fig. 6. needs to be adjusted. "Economic losses" do not belong to the environment. I don't understand why "reduction waste recycling" appears. Instead of "waste content" you should write "waste amount". Instead of "contamination of land, water and air" it should be written more precisely: "wastewater contamination with drugs". In addition, "increase in the amount of packaging waste" and "increase in the amount of medical waste" should be mentioned.

--Authors’ comments:
Thanks for this comment.  Figure has been revised accordingly.

* Conclusions must be formulated according to the authors' own results. You don't need to rely so much on other authors (with references). This is not a discussion chapter, but a conclusion chapter.

--Authors’ comments:
Thanks for this comment. Conclusion has been revised as suggested, showing and discussing main results of the paper and implications for environmental science. Moreover, the limitations has been revised also.
Thanks again.

Reviewer 2 Report

The paper is a particular and detailed presentation of the reduction of many contaminants during the months of lockdown in many countries. The variations in the decrease of contaminants are in relation to the different lockdown periods imposed by different countries.

But the conclusions are very weak, as the effects on the environment are not examined. Otherwise the study appears sterile.

In the bibliography the Rais Akthar's book on Coronavirus (COVID-19), Outbeaks, Vaccintion, Politics and Society. The countinuing challenge. Springer nature 2022.

Author Response

**The paper is a particular and detailed presentation of the reduction of many contaminants during the months of lockdown in many countries. The variations in the decrease of contaminants are in relation to the different lockdown periods imposed by different countries.
**But the conclusions are very weak, as the effects on the environment are not examined. Otherwise the study appears sterile.
--Authors’ comments:
Thanks for this comment. Conclusion has been revised as suggested, showing and discussing main results of the paper and implications for environmental science. Moreover, the limitations has been revised also.
Thanks again. 

**In the bibliography the Rais Akthar's book on Coronavirus (COVID-19), Outbeaks, Vaccintion, Politics and Society. The countinuing challenge. Springer nature 2022.

--Authors’ comments:
Thanks for this main suggestion. The book has been read and used in the text and inserted in references.
Now the study is reinforced in theoretical framework to support a better analysis and discussion. 
Thanks again.